# Designing and Developing a Vision-Based System to Investigate the Emotional Effects of News on Short Sleep at Noon: An Experimental Case Study

**DOI:** 10.3390/s23208422

**Published:** 2023-10-12

**Authors:** Ata Jahangir Moshayedi, Nafiz Md Imtiaz Uddin, Amir Sohail Khan, Jianxiong Zhu, Mehran Emadi Andani

**Affiliations:** 1School of Information Engineering, Jiangxi University of Science and Technology, Ganzhou 341000, China; ajm@jxust.edu.cn (A.J.M.); nafizmdimtiazuddin@outlook.com (N.M.I.U.); mrsohail21@gmail.com (A.S.K.); 2School of Mechanical Engineering, Southeast University, Nanjing 211189, China; 3Department of Neurosciences, Biomedicine and Movement Sciences, University of Verona, Via Casorati, 37131 Verona, Italy

**Keywords:** BlazePose, computer vision, pose detection, sleep movement, napping

## Abstract

**Background:** Sleep is a critical factor in maintaining good health, and its impact on various diseases has been recognized by scientists. Understanding sleep patterns and quality is crucial for investigating sleep-related disorders and their potential links to health conditions. The development of non-intrusive and contactless methods for analyzing sleep data is essential for accurate diagnosis and treatment. **Methods:** A novel system called the sleep visual analyzer (VSleep) was designed to analyze sleep movements and generate reports based on changes in body position angles. The system utilized camera data without requiring any physical contact with the body. A Python graphical user interface (GUI) section was developed to analyze body movements during sleep and present the data in an Excel format. To evaluate the effectiveness of the VSleep system, a case study was conducted. The participants’ movements during daytime naps were recorded. The study also examined the impact of different types of news (positive, neutral, and negative) on sleep patterns. **Results:** The system successfully detected and recorded various angles formed by participants’ bodies, providing detailed information about their sleep patterns. The results revealed distinct effects based on the news category, highlighting the potential impact of external factors on sleep quality and behaviors. **Conclusions:** The sleep visual analyzer (VSleep) demonstrated its efficacy in analyzing sleep-related data without the need for accessories. The VSleep system holds great potential for diagnosing and investigating sleep-related disorders. The proposed system is affordable, easy to use, portable, and a mobile application can be developed to perform the experiment and prepare the results.

## 1. Introduction

Sleep is a natural state of temporary unconsciousness and reduced sensory activity that occurs in animals, including humans. Sleep quality is an essential determinant of optimal physical and mental health. The restorative properties of sleep enable the body to repair and regenerate tissues, including muscle, bone, and skin [1]. Furthermore, the cerebral processes that occur during sleep, such as memory consolidation, decision making, and problem solving, are vital for cognitive function and overall well-being [2,3]. Sleep is also a critical regulator of the immune system, promoting the production of cytokines, which are instrumental in fighting infections and reducing inflammation [4]. Emotional regulation and mood enhancement are additional benefits of good sleep quality. In contrast, sub-optimal sleep quality has been linked to an elevated risk of chronic diseases such as diabetes, cardiovascular disease, and obesity [5]. In general, good sleep quality is a key factor in cognitive productivity and peak human performance, which should be an integral principle in people’s health and wellness regimes [6,7]. The quality of sleep can be influenced by a variety of factors, among which sleep position and posture play a significant role [1,2,8]. In order to effectively monitor, supervise, and analyze sleep, the implementation of appropriate designs and methodologies is indispensable. By employing suitable technologies and instruments, researchers and healthcare professionals can gather comprehensive data on sleep architecture, duration, efficiency, and the occurrence of specific sleep stages. One of the pivotal determinants that exhibits promising potential in monitoring sleep quality is the positioning adopted during sleep, which demonstrably exerts an impact on the overall quality of sleep [9]. Detection of sleep poses provides invaluable information about an individual’s posture, which can affect the quality of sleep in various ways [10]. Discomfort, pain, and snoring are common complaints associated with sleeping in inappropriate positions, which can lead to fragmented and disrupted sleep [11,12]. Moreover, movement patterns during sleep, such as tossing and turning, can also negatively impact sleep quality. Therefore, the detection of sleep poses and movements can be invaluable in providing insights into the frequency and duration of such activities, which is essential for a more comprehensive evaluation of sleep quality [13]. Therefore, it is essential to pay attention to the body’s angle during sleep and adjust it as necessary to promote optimal sleep quality and overall health [14].

Understanding sleep patterns is of paramount importance for overall health. Quality sleep serves as a vital component of physical and mental rejuvenation, fostering well-being. It significantly impacts cognitive functions, emotional stability, and mental health. Furthermore, sleep patterns play a pivotal role in maintaining physical health, including immune system function, cardiovascular well-being, and metabolic processes. Finally, they directly affect productivity, safety, and alertness, contributing to our ability to perform daily tasks effectively and preventing potential accidents.

Paying attention to the angle of the organs in humans during half-day sleep, as an essential point in order to optimize the quality of sleep and maintain proper posture, is considered the main goal of this research. An additional crucial factor to consider is the type and time of sleep, specifically circadian rhythm sleep [15]. By adhering to a regular sleep schedule that aligns with the circadian rhythm, individuals can enhance sleep quality and promote overall well-being [16]. Daytime sleep, known as napping, is a cultural practice observed in various societies globally. Some countries in Europe, the Middle East, and Africa have cultures where napping is prevalent. The practice of napping during the day is believed to improve cognitive function, productivity, and well-being and has significant cultural and social value [14]. Stress, anxiety, and other mental health conditions can interfere with sleep quality, as can certain medications and medical conditions [10]. The impact of news on human sleep has been a topic of interest in recent years as more people consume news through various media channels. It is reported that positive, negative, or neutral information can have a significant impact on sleep quality. Studies have shown that exposure to positive news before bedtime can have a calming effect on the mind and body, leading to better sleep quality and feeling more refreshed in the morning [17]. Conversely, exposure to negative news can lead to increased levels of stress and anxiety, which can interfere with sleep quality and lead to feelings of fatigue and exhaustion the next day [18]. Additionally, individual differences can also play a significant role in how news affects sleep, as some people may be more affected by negative news than others [19,20].

Recent investigations into the realm of sleep have embarked on a multifaceted exploration of human sleep patterns, underscoring the increasing significance of sleep studies in recent years. These endeavors encompass a diverse array of perspectives and methodologies, reflecting the growing recognition of the pivotal role that sleep plays in human lives. Jun et al., in their 2022 study, endeavored to procure data concerning the natural sleep patterns of individuals and the subsequent identification of distinct sleep phases. The data acquisition process encompassed an array of sleep behaviors, including nocturnal movements, snoring occurrences, and fluctuations in body temperature. In addition, the study incorporated an extensive set of environmental parameters such as ambient temperature, humidity levels, ambient luminance, carbon dioxide concentrations, and ambient acoustic conditions. These investigations were conducted on a diverse set of participants spanning a wide age range. To facilitate data collection, a technologically sophisticated smart pillow, equipped with eight pressure sensors, was employed, achieving an impressive posture discrimination accuracy range of 94–97% [21].

In the study conducted by Gaiduk et al. in 2023, their primary objective was to investigate the potential substitution of subjective sleep monitoring with data extracted from a Samsung Galaxy Watch 4, employing 166 overnight recordings as their dataset. This comprehensive inquiry centered on four critical sleep metrics, revealing that data acquired from the smartwatch exhibited a remarkable capacity to accurately estimate both the initiation and cessation of sleep, culminating in an average sleep efficiency of 89.72%. Nevertheless, a degree of variability was noted, particularly in the assessment of total sleep duration, thereby warranting prudence in contemplating its replacement. The feasibility of such substitutions hinges upon the establishment of acceptable congruence thresholds between objective and subjective measurements, a concept expounded upon in their research [22].

In parallel research conducted by Huang et al. in 2023, the introduction of the deconvolution and self-attention-based model (DCSAM) and the Gaussian noise data augmentation (GNDA) models ushered in innovative methodologies for the identification of sleep phases. This pioneering approach strategically addressed the challenges posed by skewed data distribution and minority representation of certain sleep phases, achieving an impressive accuracy rate of 90.26% and a macro F1 score of 86.51% when applied to pediatric data. Furthermore, their methodology showcased exceptional performance on the Sleep-EDFX dataset for adult subjects, thereby indicating its potential for augmenting medical applications. The central ambition of Huang et al.’s work resided in the enhancement of sleep-phase identification accuracy through the synergistic application of DCSAM with GNDA augmentation [23].

Furthermore, Boiko et al., in their 2023 study, sought to evaluate a non-invasive approach employing an accelerometer sensor for the measurement of cardiorespiratory variables during sleep. Their investigation also entailed the identification of the optimal sensor placement for precise data acquisition. Analyzing ballistocardiogram signals from 23 subjects, their findings revealed an average inaccuracy of 2.24 beats per minute (bpm) for heart rate and 1.52 respirations per minute (rpm) for respiratory rate, regardless of the sleep position. Gender-specific analysis demonstrated heart rate inaccuracies of 2.28 bpm for males and 1.41 bpm for females, and respiratory rate inaccuracies of 2.19 rpm for males and 1.30 rpm for females. These insights contribute to the understanding of the efficacy of utilizing accelerometer sensors for cardiorespiratory monitoring during sleep [24].

Lastly, in a separate 2023 investigation, Park and his colleagues introduced the customized deep sleep recommender system (CDSRS), meticulously designed to deliver personalized deep sleep services. Their methodology hinged on the application of k-means clustering to delineate distinct sleep patterns and the employment of a hybrid learning approach, seamlessly amalgamating user-based and cooperative filtering techniques. Data collection encompassed private information retrieved from handheld devices and AI motion beds, including parameters such as snoring, sleep duration, movement, and ambient noise. Noteworthy achievements were evident in CDSRS, surpassing conventional collaborative filtering (CF) and content-based filtering (CBF) models, evident through a 13.2% reduction in mean squared error (MSE) and a 14.7% enhancement in mean absolute percentage error (MAPE) in contrast to CF. This underscores the system’s efficacy in providing precise and personalized sleep recommendations, culminating in an impressive 94.2% accuracy rate [25].

According to a study by Baglioni et al. (2014), negative information, such as worrying news or stress-related information, can significantly impair sleep quality. Conversely, positive information, such as pleasant images or peaceful music, can improve sleep quality [26]. Several other studies have found similar results, suggesting that the type of information individuals receive before going to bed can have a significant impact on their sleep quality [27,28]. Understanding the effects of positive and negative reports on sleep quality can help individuals make informed decisions about their pre-sleep routines and ultimately improve their sleep quality and overall well-being. This paper aims to explore the effects of positive and negative information on sleep and provide a review of the relevant research in this area. The research contributions are listed below:Developing the sleep visual analyzer (VSleep) system for non-intrusive analysis of sleep movements using vision data.Conducting a case study on participants’ naps, revealing the impact of news categories on sleep patterns for the first time.

The authors’ research introduces a novel system for analyzing nap sleep, which is the first of its kind. This system has the potential to serve as a diagnostic tool for sleep disorders and abnormal sleep patterns. Its application could greatly contribute to the timely identification and management of related health conditions. Moreover, the proposed system offers affordability, ease of use, portability, and the capacity to develop a mobile and remote application for conducting experiments and preparing comprehensive results to analyze napping. The remaining paper is organized in the following manner: In the second section, the design aspects of the VSleep system are elaborated upon. This includes the various components of the system and the methodology used for angle calculation based on sleep movements. This section provides detailed insights into the development and implementation of the VSleep system. The third section focuses on the case study conducted to evaluate the effectiveness of the VSleep system. This section encompasses the description of the experimental protocol, including the testing conditions and the selection of participants. This section highlights the rigorous methodology employed to ensure the validity and reliability of the study’s findings. In the fourth section, the results obtained from the case study are presented and analyzed. This section includes a comprehensive analysis of the data collected during the study, which sheds light on the performance and capabilities of the VSleep system. The results are interpreted and discussed, providing valuable insights into the sleep positions, habits, and behaviors of the participants. Finally, in the fifth section, the paper concludes by summarizing the key findings and contributions of the research study. This section highlights the significance of the VSleep system in non-intrusive and contactless sleep analysis, its potential applications in diagnosing sleep-related disorders, and its role in advancing sleep medicine. The Conclusions also addresses the implications of the study’s findings for future research and developments in the field.

## 2. System Design

The VSleep design is a crucial component of this study, offering vision-based sensors and algorithms for accurate sleep posture monitoring. The proposed VSleep system, which operates on vision-based technology, does not require any physical contact with the participant. It relies solely on image data and is designed to function effectively with a camera placed at a specified distance from the participant (Figure 1A), eliminating the need for any direct physical interaction.

As mentioned earlier, vision-based systems used in sleep monitoring offer valuable capabilities, but they also face several challenges. These challenges include privacy concerns, camera placement and visibility, lighting conditions, and analysis complexity. Privacy concerns arise because video analysis requires capturing and analyzing video recordings of individuals during sleep, which may not be suitable for all participants or research settings. Additionally, ensuring the optimal camera placement without obstructing the sleep environment or causing discomfort to the user can be challenging. Adequate lighting is crucial for clear video analysis, and poor lighting can affect angle measurements, potentially disrupting the natural sleep environment. Furthermore, analyzing video recordings can be time consuming and require expertise in image processing and analysis, with the potential for subjectivity and human error in manual analysis or limitations in automated analysis methods. To address these issues, this paper proposes the use of a VSleep design, which helps alleviate the analysis complexity. Additionally, the experiment encountered fewer problems related to lighting conditions, camera placement, and visibility. Overall, this suggested method shows promise in resolving three out of the four challenges associated with vision-based sleep monitoring.

The VSleep design is a crucial component of this study, offering vision-based sensors and algorithms for accurate sleep posture monitoring. It integrates input data collection for the VSleep monitoring system, which uses a simple mobile camera as a precision sensor, collects data on body angles throughout the experiment time, and applies signal processing techniques to enhance accuracy. With its user-friendly interfaces, it provides valuable insights into sleep posture patterns and their impact on sleep quality (Figure 1).

The design system, as depicted in the block diagram in Figure 1, is made up of four parts: input data collection and capture data that can be used with any camera to capture the videos; capture data storage; the VSleep graphical user interface (GUI), which contains the VSleep GUI system, demonstrates display parts, and is used for data analysis; and output and body angle stamp in video and CSV format output.These parts are described below.

As exemplified in Figure 1, the meticulously designed GUI is adorned with five discernible buttons, each endowed with specific functionalities. The “browse” button, for instance, extends an invitation to users to traverse their computer’s repository of recorded video clips in the ubiquitous mp4 format, facilitating the discernment and selection of target files or directories. Upon the invocation of the “browse” button or option, a modal dialogue or folder dialogue elegantly materializes, thereby affording users the latitude to meticulously sift through and nominate the desired file or folder. Upon finalization of this selection process, the chosen pathway or file is ordinarily rendered manifest within a textual field or as an integral constituent of the broader user interface. Conversely, the “start” button, when activated, triggers a well-defined course of action or process. In this context, the act of “starting” encompasses the initiation of a program and the instigation of a program workflow that serves the discerning purpose of delineating landmarks within the confines of the video clip. The “pause” function, on the other hand, assumes the role of transiently suspending an ongoing process or task, ushering in a momentary hiatus to ensure precision and fine-tuning of proceedings. Moreover, the “export data” function assumes a pivotal function of both preserving and facilitating the seamless transference of data from the confines of the application to external files, seamlessly adopting the universally recognized CSV file format for maximal interoperability. Lastly, the “quit” function, of monumental import, concludes the user’s interaction with the application by orchestrating an elegant exit strategy. Prudent implementation of these multifaceted functions into the designed GUI inevitably culminates in a discernible augmentation of the application’s usability and functional prowess. This augmentation fosters a heightened level of intuitiveness, thereby endearing the VSleep application to its user base and rendering it resoundingly user friendly.

### 2.1. Input Data Collection and Captured Data for VSleep Monitoring System

The sleep system (Figure 1) analyzes video data captured by any camera and transfers the recorded data to a user’s computer. For testing, we utilized the camera of the OnePlus 9 Pro, which includes a primary 48 MP sensor with an f/1.8 aperture lens, an ultra-wide 50 MP sensor with a distinctive freeform lens design, and a 2 MP monochrome sensor. However, it is worth noting that the sleep system is compatible with various mobile phones and video-capturing devices. It should be mentioned that the height of the body should be visible in captured images. In the tested case, considering the dimensions of the bed (238.76 cm in length, 38.1 cm in height, and 125 cm in width), the distance between the camera and the bed is fixed at 100 cm at a height of 120 cm.

### 2.2. The Captured Data Storage

A Lenovo laptop was used with the following specifications.The PC configuration used for testing involved an NVIDIA GeForce GTX 1050 It graphics card paired with an Intel Core i7-8750H CPU @ 2.20 GHz. Notable features included support for up to an 8K display at 60 frames per second, a 90 mm fan design, and varying clock speeds. The graphics card had 4 GB of GDDR5 memory with a 128-bit interface, operating at a speed of 1752 MHz. The GPU ran at a base clock of 1291 MHz and was boosted up to 1392 MHz. Notably, the laptop exhibited a diminutive weight of 1.85 kg, rendering it highly portable and exceedingly convenient for utilization.

### 2.3. VSleep Graphical User Interface (VSleep GUI)

The VSleep graphical user interface (GUI) has been specifically developed for conducting human sleeping posture testing. Python (version 3.11.5), as a programming language, serves as the coding environment for designing user interfaces utilizing native elements. To facilitate the coding process, the utilization of five Python packages is imperative: OpenCV, MediaPipe, NumPy, CSV, and PyQt5. Each of these packages plays a vital role in enabling various functionalities and features of the VSleep GUI. A comprehensive overview of the responsibilities assigned to each of these packages is presented in Table 1.

The libraries mentioned in Table 1 serve as an integral component for calculating the angle and posture of the human body. At the core of the VSleep graphical user interface (GUI), the BlazePose model is employed to accurately identify and designate specific anatomical points as landmarks.

Subsequently, the angles between these landmarks are calculated, as elucidated in the subsequent sections. The angle exploration process consists of two essential elements: the landmark specifier and the angle calculation, which are comprehensively described below.

#### VSleep Landmark Specifier Using BlazePose Method

As mentioned before, the landmark specifier BlazePose method has already been used to specify body parts. This method is a computer vision technology developed by Google that uses deep learning algorithms to estimate the human body’s pose, including key points such as joints and limbs, from video or image input. This technology is based on the BlazeFace detection system and is designed to be lightweight and efficient, making it suitable for use on mobile devices. BlazePose has several applications, including fitness tracking, augmented reality, and virtual try-on technology. In fitness tracking, BlazePose can be used to monitor the accuracy of exercises, track progress over time, and provide feedback on form and technique. In augmented reality, BlazePose can enable realistic virtual avatars and virtual object interactions. In virtual try-on technology, BlazePose can be used to simulate how clothing or accessories would fit on a person’s body [29]. Previous research showed the capabilities and performance of BlazePose in different areas. Bazarvsky et al. (2020) [30] introduced BlazePose Full and BlazePose Lite models, which achieved accuracy rates of 84.1% (AR dataset) and 84.5% (Yoga dataset) for the Full AR dataset and 79.6% (Augmented Reality (AR) dataset) and 77.6% (Yoga dataset) for the Lite AR dataset, respectively. In another study, Mroz et al. (2021) [31] compared BlazePose to other models and found it to exhibit an average correlation of 0.88 compared to the Open Pose baseline. Min (2022) [32] proposed a BlazePose-based system for physical testing, achieving a motion recognition rate of 95.2% and a counting algorithm accuracy of 97.9%. Setiyadi et al. (2022) [33] developed a human activity detection system based on BlazePose, which demonstrated key point accuracy of 10–17% and achieved activity detection rates above 80%.

In the research conducted by S. Alsawadi et al. (2022) [34], a refined iteration of BlazePose was presented, incorporating supplementary edges to optimize spatial–temporal graph analysis. Noteworthy outcomes were observed across two distinct datasets: firstly, the Kinetics dataset showcased a commendable accuracy performance of 40.1%. Secondly, employing the cross-subject evaluation criteria on the NTU-RGB+D dataset, the proposed system achieved a notable accuracy metric of 87.59%. Overall, BlazePose’s accurate and efficient body pose estimation capabilities make it a valuable tool for a wide range of applications in computer vision and human–computer interactions [35]. Based on the available resources provided by the authors, the utilization of the BlazePose algorithm in sleep studies appears to be a novel approach regarding the chosen application, which involves analyzing body posture during sleep. The implementation of BlazePose is depicted in Figure 2. This state-of-the-art framework is designed to meticulously detect and track twelve distinctive landmarks on the human body. As depicted in Figure 2, the aforementioned twelve points encompass a comprehensive examination of the upper extremities, including the intricate structures of the shoulder region, elbows, wrist area, hips, knees, and ankles. Each of these points is meticulously assessed bilaterally, ensuring a comprehensive evaluation of both the right and left sides. A comprehensive elucidation of the BlazePose algorithm is provided in [30,31]. In this design, the angle calculation is conducted based on the landmark specified by the BlazePose model, which takes one picture and extracts key body points. Then, as shown in Figure 2C, each point is connected to another one, and it precisely infers two-dimensional human body landmarks from a single frame. The degree exploration consists of the following steps.

Obtain the landmarks: Use the BlazePose model to obtain a set of landmarks representing body parts. These landmarks are 2D coordinates (x, y) on the image or video frame. The normalized coordinates of various body landmarks detected by the MediaPipe library represent the position of specific body parts relative to the image frame.

Identify the landmarks: Locate the three specific landmarks as the angle exploration target, such as the shoulder, elbow, and wrist landmarks, named as three points (*A*, *B*, and *C*). Then, obtain the coordinates of the three points (*A*, *B*, and *C*) from the detected pose landmarks. Convert the landmark coordinates to pixel values. This is achieved by multiplying the normalized coordinates (ranging from 0 to 1) by the image dimensions (width and height). Let us denote the pixel coordinates as $A(X_{a},Y_{a})$, $B(X_{b},Y_{b})$, and $C(X_{c},Y_{c})$.

Calculate the vectors: Determine two vectors by subtracting the coordinates of the shoulder landmark from the elbow landmark ($\bar{AB}$), and the coordinates of the elbow landmark from the wrist landmark ($\bar{BC}$) (Equations (1) and (2)). Each vector represents the direction and magnitude between the corresponding landmarks.
(1)$$\bar{AB}=\left(X_{b}-X_{a},Y_{b}-Y_{a}\right)$$
(2)$$\bar{BC}=\left(X_{c}-X_{b},Y_{c}-Y_{b}\right)$$

**Compute the dot product of two vectors:** Calculate the dot product of the shoulder-to-elbow vector ($\bar{AB}$) and the elbow-to-wrist vector ($\bar{BC}$). The dot product measures the similarity or alignment between the vectors (Equation (Equation 3)).
(3)$$Dotproduct=X_{AB}\cdot X_{BC}+Y_{AB}\cdot Y_{BC}$$

**Compute the magnitudes of the vectors**: Find the lengths (magnitudes) of both vectors $\bar{AB}$ and $\bar{BC}$ using the Euclidean distance formula (Equations (4) and (5)).
(4)$$\left||AB|\right|=\sqrt{{X_{AB}}^{2}+{Y_{AB}}^{2}}$$
(5)$$\left||BC|\right|=\sqrt{{X_{BC}}^{2}+{Y_{BC}}^{2}}$$

**Determine the angle between two vectors**: Equation (6) can be used to calculate the angle between the two vectors (θ) in degrees using the inverse cosine function (acos).
(6)$$\theta =\arccos\left(\frac{\mathrm{Dot}\;\mathrm{Product}}{\left||AB||\times ||BC|\right|}\right)\times \frac{180}{\pi }$$

### 2.4. The VSleep GUI Parts and Sections

The VSleep design comprises five notable components, namely, browser, start, pause, export, and exit, as depicted in Table 2. These components, as labeled in Figure 1, serve as integral elements of the overall system, each fulfilling specific functions and contributing to the seamless operation of the VSleep design.

As shown in Table 2, the agreement between the GUI parts enables the design to specify and extract the angle between the human body parts at the 12 points mentioned.

Algorithm 1 outlines the whole VSleep GUI, including the menu keys and angle calculation process, into the five parts mentioned in Figure 1C. The program waits for user input to select one of the available options: browse, start, pause, export, or quit. Upon launching the application, an interface window is presented. Subsequently, the tester is able to navigate to the recorded videos by selecting the ”browse” option. Upon selecting the appropriate data clip, the program proceeds to exhibit the outcome in the form of an OpenCV window, as indicated in the code. Subsequently, the data undergoes processing within the land-marking system, which comprises twelve designated points adhering to the connections specified by the BlazePose system. Subsequently, conversion to the RGB format is necessary. Verification ensues to ensure the presence of the requisite landmarks, and the corresponding land-marking angles are copied into a CSV file. This action triggers the appearance of a data capture window, which showcases the tester’s performance. Within this window, precise observations of the tester’s movements, gestures, and other relevant actions can be made. As shown in the code in the browser part, the acquired clips are extracted inside the GUI. Once a file is selected, the program proceeds to read the frames from the file. During this step, BlazePose is implemented on the clips, and the landmarks are shown on the clips. Once a subject is within the frame, the program processes it using the MediaPipe library to extract pose landmarks. The extracted pose landmarks provide the necessary information to calculate body angles based on Equations (2)–(5). Each movement in the frame will repeat the angle calculation procedure, and then the calculated body angles can be written to a CSV file for further analysis or visualization. Additionally, the program can display the body angles on the video frame itself, creating an overlay to visualize the angles in real time. All the performance results in terms of degrees for each participant are stored in a CSV file using the append mode. At each iteration, the program checks for the ”pause” option. If the user selects ”pause”, the program pauses the capturing or processing of frames until the user resumes. Exporting the data is accomplished by clicking the ”export” button located within the controller window, whereby the displayed information is directed to the control output. In the event of program delays, an automatic termination is triggered, while the tester also possesses the option to manually cease the video playback using the “quit” function within the controller.
**Algorithm 1** The VSleep functions and human sleep posture detection
  **procedure:**
**Part one – Browse Mode**
      **Input:** Detected human body
      **Output:** Human body angle detection
      Call System File Browser() → Await File Input from User()
      **if** File received **then** Check File Alignment()
          **if** File is Aligned **then** Data Capture(), Landmark Processing(),
  Check Landmarks (All 12 Angles), Co-ordinate Conversions(), Calculate Angles (a, b, c)
             Record in CSV(), Rendering Detection()
             **if** All Pose Correct **then** Continue the Clip()
             **else**
                 Restart Rendering()
             **end if**
          **else**
             Display Hide()
          **end if**
      **else**
          Notify File Retrieval Error()
      **end if**
  **end procedure**
  **procedure:**
**Part two – Start Mode**
**(****Clip Paused:****)**
      **if** User Starts **then**, Continue the Clip()
      **else**
          Return()
      **end if**
  **end procedure**
  **procedure:**
**Part three – Pause Mode**
**(****Clip Continuing:****)**
      **if** User Pause **then**, Pause the Clip()
      **else**
          Return()
      **end if**
  **end procedure**
  **procedure:**
**Part four – Export Mode**
      **if** File is Aligned **then** OpenCV Open() → Console Output
      **else**
          Go to Browse Mode()
      **end if**
  **end procedure**
  **procedure:**
**Part five – Quit Mode**
**(****Quit Program****(****)**:**)**
      **if** User Saves Session **then** Save Session()
      **end if**
      Turn off Video Playing(), Close all Windows(), Terminate Processes(), Clean up memory()
  **end procedure**


Similarly, the program also checks for the “quit” option. If the user selects “quit”, the program terminates. The program continues executing the selected option until the user chooses to quit, at which point the program terminates. Figure 3 shows the VSleep performance on different sample data in various postures and positions and the success of landmark and angle detection on the images.

As depicted in Algorithm 1, the system is structured into five distinct components, each serving a specific function: browse mode, start mode (clip paused), pause mode (clip continuing), export mode, and quit mode (quit program). In part one (browse mode), as the initial phase, the system patiently anticipates user input, where users employ the file browser to select a file. If the chosen file adheres to the prescribed format, the algorithm orchestrates a series of operations. These include data acquisition, landmark processing, validation of 12 distinct angles, coordinate transformations, precise angle calculations, data recording within a CSV file, and rendering detection. Successful validation of all angles permits seamless video clip continuation; any discrepancies necessitate rendering re-calibration. In the unfortunate instances of missing files or retrieval errors, the display remains concealed. After transitioning to part two (start mode), user interaction assumes paramount importance as it facilitates the resumption of a paused video clip. When the user triggers the start action, the suspended clip gracefully recommences its playback; otherwise, the procedure regresses to its previous state. Within part three (pause mode), the algorithm adeptly manages user input, specifically designed to suspend the ongoing video clip playback. Initiating the pause action halts the clip’s progression, while any other input triggers the procedure to return to its prior state. The fourth segment of the algorithm (part four—export mode) revolves around data exportation tasks. Successful alignment of the chosen file with the prescribed format prompts the activation of the OpenCV library, engendering console output. Conversely, in cases of incorrect file alignment, the system gracefully reverts to the browse mode. Finally, part five (quit mode) the final phase of the algorithm accounts for user preferences concerning session preservation before program termination. Opting to preserve the session ensures its safekeeping. Subsequently, the program conducts a series of crucial tasks, including the cessation of video playback, the closure of all active windows, the termination of running processes, and meticulous memory cleanup, ultimately culminating in the program’s conclusion.

## 3. Test and Experiment

To delve into the capabilities of the VSleep design and accomplish the primary objective of this research, which pertains to calculating the angle of the human body during short daytime napping, a comprehensive GUI design was implemented. This design was applied in a case study test to acquire relevant data and facilitate further in-depth analysis.

### 3.1. Data Collection and the Protocol

The experimental study involved a cohort of 10 participants, with a specific requirement dictating that only one participant could be tested per day. Therefore, a duration of three days was allotted for each participant’s involvement. The experiment itself was scheduled to take place between 12:00 p.m. and 2:00 p.m., although most of the tests were typically completed by around 1:00 p.m.

The experimental study involved a cohort of 10 participants, with a specific requirement dictating that only one participant could be tested per day. In this case study, a cohort of ten individuals, men ranging in age from 15 to 76 years, was randomly selected and they experimented with three different conditions on three different days. Data were collected following the protocol outlined in Figure 4.

All participants voluntarily signed a self-declaration (Appendix A), providing informed consent to participate in the study and for the utilization of their data for research purposes. In this study, the samples were selected through a random process, and no consideration was given to the history or background of the selected individuals. As a result, a total of thirty samples were obtained. Figure 4 represents the test process and protocol.

The experimental protocol (Figure 4) consisted of three distinct phases: the participants were instructed to relax and reach a calm state of mind. Before the test, the participants were seated in a chair for a duration of 5–10 min, allowing them to hydrate themselves with a glass of water. Furthermore, participants were advised to refrain from consuming tea, coffee, or alcohol on the morning of the testing days. The participants were asked to come on three different days to complete the test.

On all three days, in bed and just before sleeping, the participants were asked to read a sheet of paper containing a paragraph of news containing positive, negative, or neutral information (Section A.1.1). The news was randomly distributed to each participant, accompanied by instructions not to disclose its contents to other participants. On each day, one type of information was delivered to the subject to monitor the possible effect of just one small paragraph of information on sleep. The recording of sleep videos was facilitated by the placement of cameras adjacent to the participants’ beds. Several protocols were strictly adhered to during the testing phase. Participants were kept unaware of the aim of the experiment and the type of news before going to sleep. After finishing the experiment, a series of questions pertaining to sleep evaluation were posed to the participants. Three questionnaire sheets were asked to be filled out by participants after the sleeping experiment to control the situation of the test on each day (Q1), the quality of sleep in the last month (Q2), and if the participants had any sort of sleep disorder (Q3) (Appendix A). The results showed that the situation of the test did not differ for all participants across days (conditions). Moreover, none of them had sleep disorders.

### 3.2. Statistical Analysis

To validate the analyzed data within the established framework, the resulting outcomes are depicted in Figure 5 and Figure 6 based on the designed structure. Indices were analyzed by means of non-parametric tests due to the ordinal nature of the data or the non-normal distribution of the data. The Friedman test was used to test the indices across conditions separately for each angle. Moreover, the Friedman test was used to test the indices across angles separately in each condition. Post hoc comparisons were performed using the Wilcoxon signed-rank test to compare the conditions (negative, neutral, and positive information) separately in each angle and to compare the angles separately in each condition. The Spearman correlation was applied to find a significant correlation between different indices. The effect sizes of the significant results are reported as Kendall’s W and r for Friedman and Wilcoxon signed-rank tests, respectively [35,36]. The Bonferroni correction for multiple comparisons was applied. The level of significance was set at *p* < 0.050. Data are reported and represented as mean ± standard error (SE).

### 3.3. Results

The time duration in the three conditions was not significantly different (*p* = 0.741, $\chi^{2}$(2) = 0.6, W = 0.03; negative: 463.4 ± 50.1 s, neutral: 443.2 ± 57.5 s, positive: 470.8 ± 47.2 s). Analysis of the number of changes revealed a significant effect of angle for neutral (*p* = 0.048, $\chi^{2}$(11) = 19.778, W = 0.180) and positive (*p* < 0.001, $\chi^{2}$(11) = 42.518, W = 0.387) conditions. Post hoc analysis showed that for both conditions the number of changes was lower for the right hip and left hip compared to the other angles (*p* < 0.042, z > 2.032, r > 0.64). Instead, the effect of angle was not significant for the neutral condition (*p* = 0.492). Moreover, the effect of the condition was not significant (for all angles, *p* > 0.08).

Analysis of the maximum changes revealed a significant effect of angle for negative (*p* = 0.006, $\chi^{2}$(11) = 26.308, W = 0.239), neutral (*p* < 0.001, $\chi^{2}$(11) = 42.554, W = 0.387), and positive news conditions (*p* < 0.001, $\chi^{2}$(11) = 47.662, W = 0.433). Post hoc analysis showed that, for all conditions, the maximum change was lower for the ankle (right and left) and hip (right and left) compared to the other angles (*p* < 0.047, z > 1.988, r > 0.63). Moreover, the effect of the condition was significant on the right knee and right hip (*p* < 0.049) because of the lower values in the positive news condition compared to the other conditions (*p* < 0.039, z > 2.497, r > 0.79).

The effects of maximum stable time and minimum stable time were not significant for all conditions and angles (*p* > 0.143). Analysis of the correlation revealed a significant negative correlation between body weight and the number of changes in the neutral condition for all angles (*p* < 0.022, rho = [−0.820 −0.515]). Interestingly, in the positive news condition, the correlation between body weight and the number of changes was significant just for the lower limbs (*p* < 0.012, rho = [−0.929 −0.752]). In the negative condition, the correlation between body weight and the number of changes was significant just for the upper limbs (*p* < 0.030, rho = [−0.788 −0.682]). The other correlations were not significant (*p* > 0.118 for all).

## 4. Discussion and Conclusions

Understanding and prioritizing healthy sleep patterns is crucial because sleep is the foundation upon which our physical, cognitive, emotional, and overall well-being rest. Neglecting sleep can have far-reaching consequences, while prioritizing it can lead to improved quality of life and longevity.

Identifying sleep patterns is of paramount importance for overall health. Quality sleep serves as a vital component of physical and mental rejuvenation, fostering well-being. It significantly impacts cognitive functions, emotional stability, and mental health. Furthermore, sleep patterns play a pivotal role in maintaining physical health, including immune system function, cardiovascular well-being, and metabolic processes. Finally, they directly affect productivity, safety, and alertness, contributing to our ability to perform daily tasks effectively and preventing potential accidents.

This study incorporated image analysis using BlazePose, a novel approach called VSleep for sleep analysis, alongside a set of questions focusing on individuals’ habits and experiences following exposure to three different types of news: positive, negative, and neutral. VSleep’s primary objective was to develop a portable, contactless system for analyzing daytime sleep patterns. Sleep movement analysis is used to diagnose sleep disorders, assess sleep quality, and monitor treatment efficacy. It is valuable in both clinical settings and research, aiding in understanding sleep patterns and their impact on health. Additionally, it is used in consumer sleep-tracking devices and can benefit elderly care and pediatrics.

This research work provides an overview of the formation of human body angles while lying down, elucidating the associated alterations with the aid of a computer vision system. The purpose of this study is to conduct a data analysis of the internal position (angular position) changes in the human body during sleep. The study employs a two-fold approach: firstly, to assess the functionality and performance of the proposed system in detecting the 12 key movement points of participants during daytime sleep; and secondly, to investigate the impact of daytime news exposure on sleep, offering valuable insights for psychological research and analysis.

In all image-processing-based vision systems, two critical challenges commonly emerge. The first challenge revolves around concealed points, or “hidden points”, denoting elements that elude the camera’s line of sight due to their specific positioning angles. In this design, meticulous attention has been paid to camera placement, ensuring an optimal angle that encompasses all potential human movements, especially in a supine position. However, scenarios involving extreme rotational angles, such as when one sleeps with a hand covering the face, pose a limitation for detection, a constraint not easily overcome even by vision systems boasting a substantial camera array. The second challenge pertains to low-light environments, where adequate illumination becomes indispensable. In such instances, a system equipped with infrared becomes a requisite. Regrettably, the designed system does not incorporate ultraviolet lighting due to its impact on the sampling time, a compromise that was considered carefully during the system’s development.

In the neutral condition, body weight and body movement frequency had a correlation, as reported in the results. We can interpret this as body weight having an impact on insomnia, meaning that the higher the body weight, the lower the insomnia [37]. On the other hand, insomnia is correlated with body movement frequency; the higher the insomnia, the higher the body movement frequency [38]. However, positive and negative information affects the upper and lower limbs differently, destroying this correlation. The positive information has a sort of relaxation effect on the hands [39]; consequently, the upper limbs lose the correlation between movement frequency and body weight. Instead, the negative information causes a sort of anxiety. When one suffers from anxiety, the body is filled with adrenaline, a hormone that gives the body a tremendous amount of energy and leads to physical agitation, causing the hands and legs to shake. Financial anxiety has more effects on the legs [40]; consequently, the lower limbs lose the correlation between movement frequency and body weight. Here, we presented a comprehensive computer-vision-based sleep monitoring system that is non-intrusive and non-contact, negating the need for participants to wear cameras, wearable gadgets, or any other accessories.

In summary, this research has the following objectives:Analyzing Sleep Patterns: The primary objective of this study is to analyze sleep patterns by monitoring body angle alterations during sleep using a computer vision system.Data Analysis: The study aims to conduct a detailed data analysis of changes in internal body position (angular position) during sleep.Insomnia Correlation: Investigating the correlation between body weight and body movement frequency in relation to insomnia is a key focus, aiming to understand how these factors influence sleep quality.Impact of News: The study explores how exposure to positive and negative news affects the movements of upper and lower limbs during sleep.Non-Intrusive Sleep Monitoring: Introducing a non-intrusive and non-contact computer-vision-based sleep monitoring system that eliminates the need for participants to wear any devices.Applications of the System: The research aims to identify potential applications for the proposed sleep monitoring system under different conditions.

Outcomes:The study reveals insights into the relationship between body weight, body movement frequency, and insomnia during sleep.It identifies distinct effects of positive and negative news on upper and lower limb movements during sleep.The proposed non-intrusive sleep monitoring system offers cost effectiveness, user friendliness, and portability, with the potential for mobile application development.The research emphasizes the importance of understanding sleep patterns for overall health.The study paves the way for improved diagnostics and personalized treatments for sleep-related disorders.

Moreover, we showed a possible application of the proposed system to monitor and evaluate sleep in different conditions.

Given the lack of prior knowledge regarding the participants’ medical histories, obtaining a more precise assessment of their health status would necessitate consultation with a medical professional. This step is delineated as part of the subsequent phases of this research and is a focus of future work. We should mention that the participants did not report any history of mental or neurophysiological disease. Moreover, the scores of the questionnaires confirmed that they had no sleep disorder or problem. Future studies with a higher number of participants and different conditions can be performed with the proposed system. The findings of the case study emphasized the importance of understanding sleep patterns and their impact on health. The comprehensive analysis provided by the VSleep system contributes to advancing knowledge in the field of sleep medicine. Further research and development of non-intrusive sleep analysis methods hold promise for improving diagnostic techniques and personalized treatment approaches for sleep-related disorders. The sleep visual analyzer (VSleep) demonstrated its efficacy in analyzing sleep-related data without the need for accessories. Therefore, the VSleep system opens a new door to easier diagnosis and the investigation of sleep-related disorders. The proposed system is cost effective, user friendly, portable, and has the capability to create a mobile application for conducting experiments and generating results.

The low number of subjects is one of the limitations of this study. In future work, with a high number of subjects, in addition to having more reliable results, it will be possible to investigate the effects of gender (male, female), age (young, middle-aged, old), and education (high school, university).

## Figures and Tables

**Figure 1 sensors-23-08422-f001:**
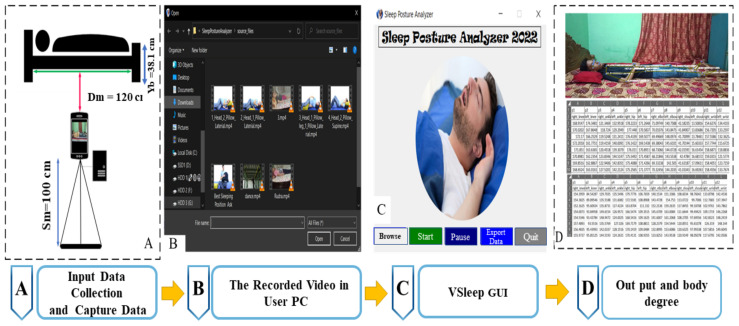
VSleep block diagram and parts. (**A**) Input data collection and capture; (**B**) capture data storage; (**C**) the VSleep graphical user interface (GUI); and (**D**) output and body angle stamp on video and CSV format output.

**Figure 2 sensors-23-08422-f002:**
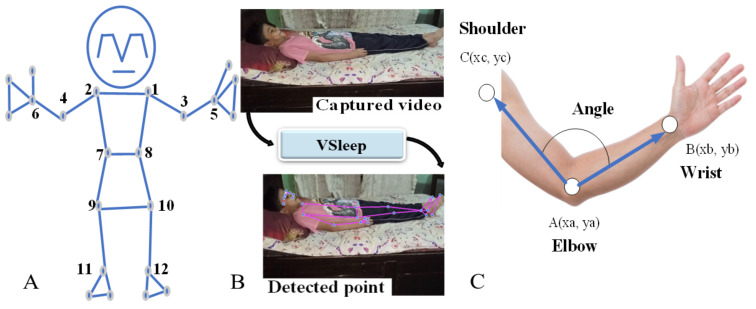
BlazePose topology and performance. (**A**) The twelve detected points on a human body (1. left shoulder, 2. right shoulder, 3. left elbow, 4. right elbow, 5. left wrist, 6. right wrist, 7. left hip, 8. right hip, 9. left knee, 10. right knee, 11. left ankle, 12. right ankle). (**B**) The VSleep design detected point. (**C**) One sample of angle measurement with VSleep.

**Figure 3 sensors-23-08422-f003:**
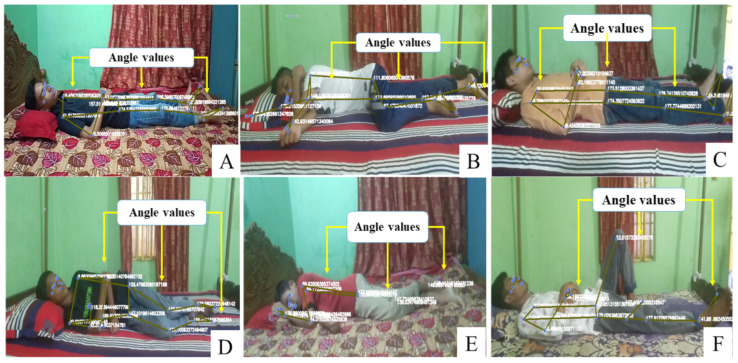
VSleep system’s performance in angle detection across participants in different sleep positions, including those on their back (**A**,**C**,**F**), right side (**B**,**D**,**E**), is illustrated by blue points indicating successful detection points.

**Figure 4 sensors-23-08422-f004:**
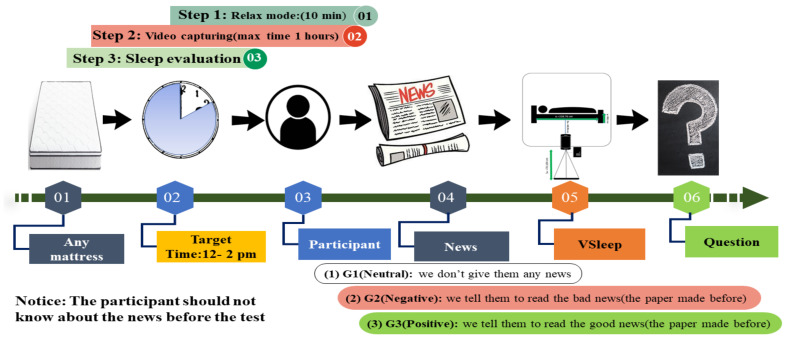
The experimental test protocol.

**Figure 5 sensors-23-08422-f005:**
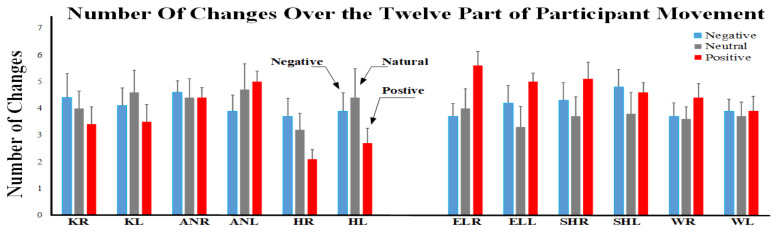
Number of changes in three conditions for all angles. KR: knee right, KL: knee left, ANR: ankle right, ANL: ankle left, HR: hip right, HL: hip left, ELR: elbow right, ELL: elbow left, SHR: shoulder right, SHL: shoulder left, WR: wrist right, WL: wrist left.

**Figure 6 sensors-23-08422-f006:**
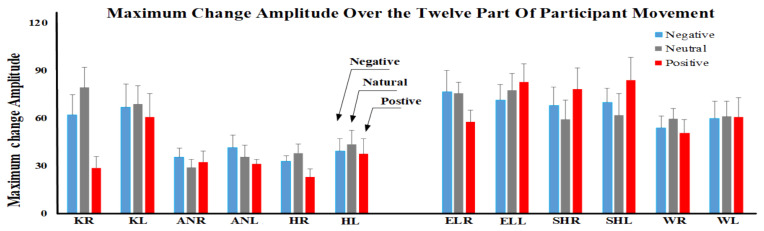
Maximum angle changes in three conditions for all angles. KR: knee right, KL: knee left, ANR: ankle right, ANL: ankle left, HR: hip right, HL: hip left, ELR: elbow right, ELL: elbow left, SHR: shoulder right, SHL: shoulder left, WR: wrist right, WL: wrist left.

**Table 1 sensors-23-08422-t001:** GUI packages used in VSleep.

Library Name	Aim and Duty
OpenCV	**Aim:** Empower computer vision applications with an efficient, open-source framework.
	**Duty:** To encapsulate, the process involves acquiring data, detecting object pose, and displaying values during object motion.
MediaPipe	**Aim:** Pose estimation
	**Duty:** Execute the BlazePose model to identify landmarks, utilizing its versatile, cross-platform framework renowned for providing robust, pre-constructed solutions tailored to computer vision tasks.
NumPy	**Aim:** Degree calculation
	**Duty:** Calculate object degrees by using the cosine function.
CSV	**Aim:** Export the data
	**Duty:** Record the angular displacements of the object’s movements within a CSV file.
PyQt5	**Aim:** Enables the use of the Qt framework and graphical user interfaces
	**Duty:** Facilitate diverse capabilities such as data export, pause, and system exit options.

**Table 2 sensors-23-08422-t002:** The VSleep graphical user interface (GUI) sections and parts.

Label: Name	Purpose and Duty
A: Load	Purpose: Loading the captured video.
	Duty: When this option is clicked, the data-capturing window will appear.
B: Performance	Purpose: Performance monitoring.
	Duty: After selecting the ’A’ option, the performance will be displayed.
C: Pause	Purpose: To pause the system.
	Duty: Whilst the performance output persists on the screen, it is feasible to interrupt it using the designated ’C’ button if a pause is requisite.
D: Save	Purpose: Data storage.
	Duty: Within this GUI system, data are accumulated through an appending process, maintaining this mode when engaged within the OpenCV window environment.
E: Exit	Purpose: Exiting the GUI.
	Duty: In the event that the evaluator desires to terminate the program, they can simply click on this button to effectuate the program’s closure.

## Data Availability

The data are available upon request.

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
