# Peer review of "Designing and Developing a Vision-Based System to Investigate the Emotional Effects of News on Short Sleep at Noon: An Experimental Case Study"

_sensors, 2023, doi:10.3390/s23208422_

Round 1

Reviewer 1 Report

 As a reviewer, I find this paper to be interesting. However, there are minor corrections that need to be made in order for it to be suitable for publication in SENSORS. The requested revisions are as follows:

Please clarify the primary objective of the Sleep Visual Analyzer (VSleep) system.

Provide a more detailed explanation of how the VSleep system analyzes sleep movements and generates reports.

State whether physical accessories or contact with the body were required for the functioning of the VSleep system.

Specify the data format used to present the analyzed body movements during sleep.

Elaborate on the purpose of the case study conducted to evaluate the VSleep system.

Provide a comprehensive summary of the main findings and outcomes of the case study, particularly regarding sleep patterns and the impact of different types of news on sleep quality.

Once these revisions have been made, the paper will be suitable for publication in SENSORS.

 As a reviewer, I find this paper to be interesting. However, there are minor corrections that need to be made in order for it to be suitable for publication in SENSORS. The requested revisions are as follows:

Please clarify the primary objective of the Sleep Visual Analyzer (VSleep) system.

Provide a more detailed explanation of how the VSleep system analyzes sleep movements and generates reports.

State whether physical accessories or contact with the body were required for the functioning of the VSleep system.

Specify the data format used to present the analyzed body movements during sleep.

Elaborate on the purpose of the case study conducted to evaluate the VSleep system.

Provide a comprehensive summary of the main findings and outcomes of the case study, particularly regarding sleep patterns and the impact of different types of news on sleep quality.

Once these revisions have been made, the paper will be suitable for publication in SENSORS.

Author Response

Dr Ata Jahangir Moshayedi, PhD, Associate Professor
School of Information Engineering, JXUST
86 Hongqi Avenue, Ganzhou, JX, P.R.C 341000

Email: ajm@jxust.edu.cn, moshaydi@gmail.com

To

The Editor-in-Chief:

Detail answer for Response to the Reviewer(s)

Sensor , MDPI

Manuscript ID:  sensors-2619444

Paper Title:

Designing and developing a Vision-Based System to Investigate the
Emotional Effects of News on Short Sleep at Noon: An Experimental Case Study

Authors:

Ata Jahangir Moshayedi *, Nafiz Md Imtiaz Uddin, Amir Sohail Khan,Jianxiong Zhu *, Mehran Emadi Andani.

Dear Respected Editor,

The authors wish to express their gratitude to the editor and the reviewers for their meticulous review of our manuscript and for their valuable comments and suggestions aimed at enhancing the manuscript's quality. We have diligently addressed each comment in a point-by-point manner, resulting in the necessary modifications to the manuscript. The detailed corrections have been provided below, with all changes clearly highlighted in the text two different  color of read and blue.We hope that these revisions meet the expectations of the esteemed editor.

Best regards,

Ata Jahangir Moshayedi

Comment 1:

Q1. Please clarify the primary objective of the Sleep Visual Analyzer (VSleep) system.

Author’s response: Thank you for your comment. Vsleep's primary objective was to develop a portable, contactless system for analyzing daytime sleep patterns. This study incorporated image analysis using Blazepose, a novel approach to sleep analysis, alongside a set of questions focusing on individuals' habits and experiences following exposure to three different types of news: positive, negative, and neutral. Sleep movement analysis is used to diagnose sleep disorders, assess sleep quality, and monitor treatment efficacy. It's valuable in both clinical settings and research, aiding in understanding sleep patterns and their impact on health. Additionally, it's used in consumer sleep-tracking devices and can benefit elderly care and pediatrics.

The mentioned points are specied in bellow part

  1. Abstartct section 1, section 2, section 4
  2. The mentioned text add in the discussion section

Vsleep's primary objective was to develop a portable, contactless system for analyzing daytime sleep patterns. This study incorporated image analysis using Blazepose, a novel approach to sleep analysis, alongside a set of questions focusing on individuals' habits and experiences following exposure to three different types of news: positive, negative, and neutral. Sleep movement analysis is used to diagnose sleep disorders, assess sleep quality, and monitor treatment efficacy. It's valuable in both clinical settings and research, aiding in understanding sleep patterns and their impact on health. Additionally, it's used in consumer sleep-tracking devices and can benefit elderly care and pediatrics.

Q2. Provide a more detailed explanation of how the VSleep system analyzes sleep movements and generates reports.

Authors’ response:

Thank you for giving us this opportunity to explain this. The VSleep system employs the Blazepose method, utilizing a deep neural network to detect key body landmarks from input images or frames. These landmarks are then used to calculate spatial relationships and angles, as described in equations 1 to 6, enabling the estimation of a person's body pose. This process involves three stages: landmark acquisition, identification, and vector calculation. The GUI assists in collecting data from 12 key points, including Right Knee Angle, Left Knee Angle, Right Ankle Angle, Left Ankle Angle, Right Hip Angle, Left Hip Angle, Right Elbow Angle, Left Elbow Angle, Right Shoulder Angle, Left Shoulder Angle, Right Wrist Angle, and Left Wrist Angle, and storing them in a CSV format. Each CSV entry represents a unique video frame, creating a chronological sequence of body movements, while columns within the file capture specific attributes related to these 12 key points. As video playback continues, the CSV file accumulates data, offering insights into the evolving dynamics of body movements over time. As the main blaze poses described in other papers, we didn’t add more detail  about image conversion, and the reader can read more detail from [25] , and all processes of Vsleep are shown in Algorithm 1.

The mentioned points are add after the algorithm section .

 As depicted in Algorithm 1, the system is structured into five distinct components, each serving a specific function: BROWSE MODE, START MODE (CLIP PAUSED), PAUSE MODE (CLIP CONTINUING), EXPORT MODE, and QUIT MODE (QUIT PROGRAM()).

  • PART ONE – BROWSE MODE: In this initial phase, the system patiently anticipates user input, where users employ the file browser to select a file. If the chosen file adheres to the prescribed format, the algorithm orchestrates a series of operations. These include data acquisition, landmark processing, validation of 12 distinct angles, coordinate transformations, precise angle calculations, data recording within a CSV file, and rendering detection. Successful validation of all angles permits seamless video clip continuation; any discrepancies necessitate rendering recalibration. In the unfortunate instances of missing files or retrieval errors, the display remains concealed.
  • PART TWO – START MODE (CLIP PAUSED): Transitioning to this stage, user interaction assumes paramount importance as it facilitates the resumption of a paused video clip. When the user triggers the start action, the suspended clip gracefully recommences its playback; otherwise, the procedure regresses to its previous state.
  • PART THREE – PAUSE MODE (CLIP CONTINUING): Within this phase, the algorithm adeptly manages user input, specifically designed to suspend the ongoing video clip playback. Initiating the pause action halts the clip's progression, while any other input triggers the procedure to return to its prior state.
  • PART FOUR – EXPORT MODE: The fourth segment of the algorithm revolves around data exportation tasks. Successful alignment of the chosen file with the prescribed format prompts the activation of the OpenCV library, engendering console output. Conversely, in cases of incorrect file alignment, the system gracefully reverts to the browse mode.
  • PART FIVE – QUIT MODE (QUIT PROGRAM()): Finally, the ultimate phase of the algorithm accounts for user preferences concerning session preservation before program termination. Opting to preserve the session ensures its safekeeping. Subsequently, the program conducts a series of crucial tasks, including the cessation of video playback, the closure of all active windows, the termination of running processes, and meticulous memory cleanup, ultimately culminating in the program's conclusion.

Q3. State whether physical accessories or contact with the body were required for the functioning of the VSleep system.

Authors’ response: Thank you for your comment. The proposed Vsleep system, which operates on vision-based technology, does not require any physical contact with the participant. It relies solely on image data and is designed to function effectively with a camera placed at a specified distance from the participant, eliminating the need for any direct physical interaction.The mentioned points are add after  in the System design.

The VSleep design is a crucial component of this study, offering vision-based sensors and algorithms for accurate sleep posture monitoring. The proposed Vsleep system, which operates on vision-based technology, does not require any physical contact with the participant. It relies solely on image data and is designed to function effectively with a camera placed at a specified distance from the participant( Fig 1_A), eliminating the need for any direct physical interaction.

Q4. Specify the data format used to present the analyzed body movements during sleep.

Authors’ response: Thank you for your comment. Our Vsleep system utilizes commonly used mobile devices to capture input in the form of MP4 videos. The resulting data is then stored in CSV-format files for convenient output and analysis. The mentioned points are specied in the add part for the Question 2 (PART ONE – BROWSE MODE) of algorithm. 

Q5. Elaborate on the purpose of the case study conducted to evaluate the VSleep system.

Authors’ response: Thank you for your comment. The study employs a two-fold approach: firstly, to assess the functionality and performance of the proposed system in detecting the 12 key movement points of participants during daytime sleep, and secondly, to investigate the impact of daytime news exposure on sleep, offering valuable insights for psychological research and analysis.The mentioned points are add in the decision section, 

The purpose of this study is to conduct a data analysis of the internal position(angular position) changes in the human body during sleep.The study employs a two-fold approach: firstly, to assess the functionality and performance of the proposed system in detecting the 12 key movement points of participants during daytime sleep, and secondly, to investigate the impact of daytime news exposure on sleep, offering valuable insights for psychological research and analysis

Q6. Provide a comprehensive summary of the main findings and outcomes of the case study, particularly regarding sleep patterns and the impact of different types of news on sleep quality.

Authors’ response: Thank you for your valuable comment. This research shows the performance of a designed system named Vsleep. The study introduces a non-intrusive and non-contact computer vision-based sleep monitoring system, eliminating the need for participants to wear cameras or wearable devices. It suggests potential applications for monitoring sleep in various conditions and calls for future research with a larger participant pool to investigate gender, age, and education effects on sleep patterns. The VSleep system offers a cost-effective and user-friendly approach to sleep analysis, paving the way for improved diagnostics and personalized treatments for sleep-related disorders. The case study provides a comprehensive analysis of sleep patterns and their correlation with body angle alterations during sleep using a computer vision system. It explores how factors like body weight and body movement frequency relate to insomnia. Notably, it identifies that higher body weight is associated with lower insomnia, while increased insomnia corresponds to higher body movement frequency. However, the impact of positive and negative news on the upper and lower limbs disrupts this correlation.

In summary :

Objectives:

  • Analyzing Sleep Patterns: The primary objective of this study is to analyze sleep patterns by monitoring body angle alterations during sleep using a computer vision system.
  • Data Analysis: The study aims to conduct a detailed data analysis of changes in internal body position (angular position) during sleep.
  • Insomnia Correlation: Investigating the correlation between body weight and body movement frequency in relation to insomnia is a key focus, aiming to understand how these factors influence sleep quality.
  • Impact of News: The study explores how exposure to positive and negative news affects the movements of upper and lower limbs during sleep.
  • Non-Intrusive Sleep Monitoring: Introducing a non-intrusive and non-contact computer vision-based sleep monitoring system that eliminates the need for participants to wear any devices.
  • Applications of the System: The research aims to identify potential applications for the proposed sleep monitoring system under different conditions.

Outcomes:

  • The study reveals insights into the relationship between body weight, body movement frequency, and insomnia during sleep.
  • It identifies distinct effects of positive and negative news on upper and lower limb movements during sleep.
  • The proposed non-intrusive sleep monitoring system offers cost-effectiveness, user-friendliness, and portability, with the potential for mobile application development.
  • The research emphasizes the importance of understanding sleep patterns for overall health.
  • The study paves the way for improved diagnostics and personalized treatments for sleep-related disorders.

The mentioned text is added in the introduction and dissection section.

Section 1 in introduction : Understanding sleep patterns is of paramount importance for overall health. Quality sleep serves as a vital component of physical and mental rejuvenation, fostering well-being. It significantly impacts cognitive functions, emotional stability, and mental health. Furthermore, sleep patterns play a pivotal role in maintaining physical health, including immune system function, cardiovascular well-being, and metabolic processes. Finally, they directly affect productivity, safety, and alertness, contributing to our ability to perform daily tasks effectively and preventing potential accidents.

Section 2 in discussion : Understanding and prioritizing healthy sleep patterns is crucial because sleep is the foundation upon which our physical, cognitive, emotional, and overall well-being rest. Neglecting sleep can have far-reaching consequences, while prioritizing it can lead to improved quality of life and longevity.The descion section  before second paraghraph:

In summary :

Objectives:

  • Analyzing Sleep Patterns: The primary objective of this study is to analyze sleep patterns by monitoring body angle alterations during sleep using a computer vision system.
  • Data Analysis: The study aims to conduct a detailed data analysis of changes in internal body position (angular position) during sleep.
  • Insomnia Correlation: Investigating the correlation between body weight and body movement frequency in relation to insomnia is a key focus, aiming to understand how these factors influence sleep quality.
  • Impact of News: The study explores how exposure to positive and negative news affects the movements of upper and lower limbs during sleep.
  • Non-Intrusive Sleep Monitoring: Introducing a non-intrusive and non-contact computer vision-based sleep monitoring system that eliminates the need for participants to wear any devices.
  • Applications of the System: The research aims to identify potential applications for the proposed sleep monitoring system under different conditions.

Outcomes:

  • The study reveals insights into the relationship between body weight, body movement frequency, and insomnia during sleep.
  • It identifies distinct effects of positive and negative news on upper and lower limb movements during sleep.
  • The proposed non-intrusive sleep monitoring system offers cost-effectiveness, user-friendliness, and portability, with the potential for mobile application development.
  • The research emphasizes the importance of understanding sleep patterns for overall health.
  • The study paves the way for improved diagnostics and personalized treatments for sleep-related disorders.

The authors would like to express their sincere appreciation to the reviewer for their thorough consideration of the manuscript and for providing valuable feedback, which undoubtedly enhanced its quality. Building upon these insightful comments, we have revised the article accordingly. We hope that the revised paper meets the expectations of the esteemed reviewer.

We extend our best wishes to you.

 Warm regards

Authors of  sensors-2619444

Reviewer 2 Report

Journal-Access: Sensors

Designing and developing a Vision-Based System to Investigate the Emotional Effects of News on Short Sleep at Noon: An Experimental Case Study

Paper Summary:

The article presents the Sleep Visual Analyzer (VSleep), a novel system designed for non-intrusive analysis of sleep patterns using camera data and a Python GUI. This system detects body movement angles during sleep without physical contact, offering detailed sleep pattern insights. A case study demonstrates its effectiveness, revealing the impact of different news categories on sleep quality.

In general, the work is interesting and this study has merit for publication. However, there are some problems that should be addressed.

1- The introduction section can be cited with some papers published in the Journal of Sensors.

2- Paper writing method and Quality should be revised carefully.

3- Figures:

·         Text inside the figures is blurry, make them clear and bigger for all figures especially Fig. 3.

·         The image resolution for some of them is low, and the authors should improve it.

4- Could the authors clarify the specific emotional effects of news that this study investigates during short noon sleep? Also, could you provide more information about the case study mentioned? How many participants were involved? Could the authors elaborate on the selection criteria for participants in the case study, such as age, gender, or any specific sleep-related conditions?

5- In the article, it highlights the importance of sleep for overall health. Could the authors specify how this research contributes to improving sleep quality and health? What kind of healthy issues do they mean? How many participants were in healthy and how many of them were in unhealthy situation?

6- The authors mention the impact of sleep positions on sleep quality. Could they elaborate on how the VSleep system addresses this issue? Regarding the impact of news on sleep, how do they plan to explore the impact of news on sleep patterns in your research?

7- In the system design section, the authors discuss the challenges of vision-based sleep monitoring. Could you explain how the VSleep system addresses these challenges? Could you elaborate on the testing conditions? How user-friendly is the VSleep GUI, and what functionalities does it offer to users?

Paper writing method and Quality should be revised carefully.

Author Response

Dr Ata Jahangir Moshayedi, PhD, Associate Professor
School of Information Engineering, JXUST
86 Hongqi Avenue, Ganzhou, JX, P.R.C 341000

Email: ajm@jxust.edu.cn, moshaydi@gmail.com

To

The Editor-in-Chief:

Detail answer for Response to the Reviewer(s)

Sensor , MDPI

Manuscript ID:  sensors-2619444

Paper Title:

Designing and developing a Vision-Based System to Investigate the
Emotional Effects of News on Short Sleep at Noon: An Experimental Case Study

Authors:

Ata Jahangir Moshayedi *, Nafiz Md Imtiaz Uddin, Amir Sohail Khan,Jianxiong Zhu *, Mehran Emadi Andani.

Dear Respected Editor,

The authors wish to express their gratitude to the editor and the reviewers for their meticulous review of our manuscript and for their valuable comments and suggestions aimed at enhancing the manuscript's quality. We have diligently addressed each comment in a point-by-point manner, resulting in the necessary modifications to the manuscript. The detailed corrections have been provided below, with all changes clearly highlighted in the text two different  color of read and blue.We hope that these revisions meet the expectations of the esteemed editor.

Best regards,

Ata Jahangir Moshayedi

Comment 2:

Designing and developing a Vision-Based System to Investigate the Emotional Effects of News on Short Sleep at Noon: An Experimental Case Study

Paper Summary:

 The article presents the Sleep Visual Analyzer (VSleep), a novel system designed for non-intrusive analysis of sleep patterns using camera data and a Python GUI. This system detects body movement angles during sleep without physical contact, offering detailed sleep pattern insights. A case study demonstrates its effectiveness, revealing the impact of different news categories on sleep quality. In general, the work is interesting and this study has merit for publication. However, there are some problems that should be addressed.

Q1: The introduction section can be cited with some papers published in the Journal of Sensors.

Authors’ response: Thank you for your comment. As part of our review process, we have recently evaluated five new papers published in the Sensor journal, with a focus on those published in the years 2022 and 2023. We have incorporated relevant insights and findings from these papers into the introduction section of the research paper.

The bellow  text is added in the introduction section:

Recent investigations into the realm of sleep have embarked on a multifaceted exploration of human sleep patterns, underscoring the increasing significance of sleep studies in recent years. These endeavors encompass a diverse array of perspectives and methodologies, reflecting the growing recognition of the pivotal role that sleep plays in human lives.

Jun et al., in their 2022 study, endeavored to procure data concerning the natural sleep patterns of individuals and the subsequent identification of distinct sleep phases. The data acquisition process encompassed an array of sleep behaviors, including nocturnal movements, snoring occurrences, and fluctuations in body temperature. In addition, the study incorporated an extensive set of environmental parameters such as ambient temperature, humidity levels, ambient luminance, carbon dioxide concentrations, and ambient acoustic conditions. These investigations were conducted on a diverse set of participants spanning a wide age range. To facilitate data collection, a technologically sophisticated smart pillow, equipped with eight pressure sensors, was employed, achieving an impressive posture discrimination accuracy range of 94–97% [Jun, W. H., Kim, H. J., & Hong, Y. S. (2022). Sleep pattern analysis in unconstrained and unconscious state. Sensors, 22(23), 9296].

In the study conducted by Gaiduk et al. in 2023, their primary objective was to investigate the potential substitution of subjective sleep monitoring with data extracted from a Samsung Galaxy Watch 4, employing 166 overnight recordings as their dataset. This comprehensive inquiry centered on four critical sleep metrics, revealing that data acquired from the smartwatch exhibited a remarkable capacity to accurately estimate both the initiation and cessation of sleep, culminating in an average sleep efficiency of 89.72%. Nevertheless, a degree of variability was noted, particularly in the assessment of total sleep duration, thereby warranting prudence in contemplating its replacement. The feasibility of such substitutions hinges upon the establishment of acceptable congruence thresholds between objective and subjective measurements, a concept expounded upon in their research [Gaiduk, M., Seepold, R., Martínez Madrid, N., & Ortega, J. A. (2023). Assessing the feasibility of replacing subjective questionnaire-based sleep measurement with an objective approach using a Smartwatch. Sensors, 23(13), 6145].

In parallel research conducted by Huang et al. in 2023, the introduction of the DeConvolution- and Self-Attention-based Model (DCSAM) and the Gaussian Noise Data Augmentation (GNDA) models ushered in innovative methodologies for the identification of sleep phases. This pioneering approach strategically addressed the challenges posed by skewed data distribution and minority representation of certain sleep phases, achieving an impressive accuracy rate of 90.26% and a macro F1-score of 86.51% when applied to pediatric data. Furthermore, their methodology showcased exceptional performance on the Sleep-EDFX dataset for adult subjects, thereby indicating its potential for augmenting medical applications. The central ambition of Huang et al.'s work resided in the enhancement of sleep phase identification accuracy through the synergistic application of DCSAM with GNDA augmentation [Huang, X., Shirahama, K., Irshad, M. T., Nisar, M. A., Piet, A., & Grzegorzek, M. (2023). Sleep Stage Classification in Children Using Self-Attention and Gaussian Noise Data Augmentation. Sensors, 23(7), 3446].

Furthermore, Boiko et al., in their 2023 study, sought to evaluate a non-invasive approach employing an accelerometer sensor for the measurement of cardiorespiratory variables during sleep. Their investigation also entailed the identification of the optimal sensor placement for precise data acquisition. Analyzing ballistocardiogram signals from 23 subjects, their findings revealed an average inaccuracy of 2.24 beats per minute (bpm) for heart rate and 1.52 respirations per minute (rpm) for respiratory rate, regardless of the sleep position. Gender-specific analysis demonstrated heart rate inaccuracies of 2.28 bpm for males and 1.41 bpm for females, and respiratory rate inaccuracies of 2.19 rpm for males and 1.30 rpm for females. These insights contribute to the understanding of the efficacy of utilizing accelerometer sensors for cardiorespiratory monitoring during sleep [Boiko, A., Gaiduk, M., Scherz, W. D., Gentili, A., Conti, M., Orcioni, S., & Seepold, R. (2023). Monitoring of cardiorespiratory parameters during sleep using a special holder for the accelerometer sensor. Sensors, 23(11), 5351].

Lastly, in a separate 2023 investigation, Park and his colleagues introduced the Customized Deep Sleep Recommender System (CDSRS), meticulously designed to deliver personalized deep sleep services. Their methodology hinged on the application of K-means clustering to delineate distinct sleep patterns and the employment of a hybrid learning approach, seamlessly amalgamating user-based and cooperative filtering techniques. Data collection encompassed private information retrieved from handheld devices and AI motion beds, including parameters such as snoring, sleep duration, movement, and ambient noise. Noteworthy achievements were evident in CDSRS, surpassing conventional collaborative filtering (CF) and content-based filtering (CBF) models, evident through a 13.2% reduction in mean squared error (MSE) and a 14.7% enhancement in mean absolute percentage error (MAPE) in contrast to CF. This underscores the system's efficacy in providing precise and personalized sleep recommendations, culminating in an impressive 94.2% accuracy rate [Park, J. H., & Lee, J. D. (2023). A Customized Deep Sleep Recommender System Using Hybrid Deep Learning. Sensors, 23(15), 6670].

Q2: Paper writing method and Quality should be revised carefully.

Author response:Thanks for the insightful guidance. We assessed the paper's writing format and quality in latex, aligning it with the standards set by the Sensors journal. Necessary adjustments were made to ensure compliance with the journal's formatting requirements.

Q3: Figures:

 Text inside the figures is blurry, make them clear and bigger for all figures especially Fig. 3.

The image resolution for some of them is low, and the authors should improve it.

 Author response: Thank you for your comment. All figures have been presented with an increased text size. However, it is important to note that the data within Figure 3, as generated by Vsleep, retains its high precision and cannot be altered. The remaining aspects, including figure labels and quality, have been meticulously aligned with the requirements stipulated by the journal.

 Q4: Q4_1: Could the authors clarify the specific emotional effects of news that this study investigates during short noon sleep? Q4_2: Also, could you provide more information about the case study mentioned? How many participants were involved? Could the authors elaborate on the selection criteria for participants in the case study, such as age, gender, or any specific sleep-related conditions?

Author response: Thank you for the comment.

Q4_1: As the research outcome shows:Positive and negative information exert varying effects on the upper and lower limbs, thereby disrupting this correlation. Positive information tends to induce a relaxation effect on the hands, leading to the dissociation of the upper limbs from the relationship between movement frequency and body weight. Conversely, exposure to negative information tends to evoke feelings of anxiety. This anxiety response triggers the release of adrenaline, a hormone that imparts an abundance of energy to the body, resulting in physical restlessness. Consequently, the hands and legs exhibit tremors. Financial anxiety, in particular, exerts a more pronounced impact on the lower limbs, further attenuating the correlation between movement frequency and body weight in the lower extremities.

The mentioned points are specied in discussion section with blue color text  :However, positive and negative .. to end

Q4_2: In this case study, a cohort of ten individuals, men ranging in age from 15 to 76 years, was randomly selected. Data were collected following the protocol outlined in Figure 4. All participants voluntarily signed a self-declaration (Appendix I), providing informed consent to participate in the study and for the utilization of their data for research purposes. In this study, the samples were selected through a random process, and no consideration was given to the history or background of the selected individuals. It is added to the text before figure 4.

The mentioned points are specied in text:The experimental study involved a cohort of 10 participants, with a specific requirement dictating that only one participant could be tested per day. In this case study, a cohort of ten individuals, men ranging in age from 15 to 76 years, was randomly selected. Data were collected following the protocol outlined in Figure 4. All participants voluntarily signed a self-declaration (Appendix I), providing informed consent to participate in the study and for the utilization of their data for research purposes. In this study, the samples were selected through a random process, and no consideration was given to the history or background of the selected individuals.

Q5- In the article, it highlights the importance of sleep for overall health. Could the authors specify how this research contributes to improving sleep quality and health? What kind of healthy issues do they mean? How many participants were in healthy and how many of them were in unhealthy situations?

Author response: Thank you for your comment.Understanding sleep patterns is of paramount importance for overall health. Quality sleep serves as a vital component of physical and mental rejuvenation, fostering well-being. It significantly impacts cognitive functions, emotional stability, and mental health. Furthermore, sleep patterns play a pivotal role in maintaining physical health, including immune system function, cardiovascular well-being, and metabolic processes. Finally, they directly affect productivity, safety, and alertness, contributing to our ability to perform daily tasks effectively and preventing potential accidents. Given the lack of prior knowledge regarding the participants' medical histories, obtaining a more precise assessment of their health status would necessitate consultation with a medical professional. This step is delineated as part of the subsequent phases of this research and is a focus of future work. This point is highlighted in the discussion section.We should mention that the participants did not report any history of mental or neurophysiological disease. Moreover, the scores of the questionnaires confirmed that they had no sleep disorder or problem.The Results and discussion section were revised to mention the points. The mentioned points are specied in the discussion section.

Section 1: Figureing out the  sleep patterns is of paramount importance for overall health. Quality sleep serves as a vital component of physical and mental rejuvenation, fostering well-being. It significantly impacts cognitive functions, emotional stability, and mental health. Furthermore, sleep patterns play a pivotal role in maintaining physical health, including immune system function, cardiovascular well-being, and metabolic processes. Finally, they directly affect productivity, safety, and alertness, contributing to our ability to perform daily tasks effectively and preventing potential accidents.

Section2:  Given the lack of prior knowledge regarding the participants' medical histories, obtaining a more precise assessment of their health status would necessitate consultation with a medical professional. This step is delineated as part of the subsequent phases of this research and is a focus of future work..We should mention that the participants did not report any history of mental or neurophysiological disease. Moreover, the scores of the questionnaires confirmed that they had no sleep disorder or problem.

Q6- The authors mention the impact of sleep positions on sleep quality. Could they elaborate on how the VSleep system addresses this issue? Regarding the impact of news on sleep, how do they plan to explore the impact of news on sleep patterns in your research?

Author response: Thank you for your comment. In this research, our focus was exclusively on the extraction of movements and angles between various parts of the human body during sleep. Posture analysis was not within the scope of this study, although it is worth noting that the designed system possesses the capability to identify human postures during sleep. The analysis did not encompass an examination of sleep patterns under various news stimuli.

Q7- In the system design section, the authors discuss the challenges of vision-based sleep monitoring. Could you explain how the VSleep system addresses these challenges? Could you elaborate on the testing conditions? How user-friendly is the VSleep GUI, and what functionalities does it offer to users?

Author response:

Q7_1: Thank you for your comment, In all image processing-based vision systems, two critical challenges commonly emerge. The first challenge revolves around concealed points, or "hidden points," denoting elements that elude the camera's line of sight due to their specific positioning angles. In this design, meticulous attention has been paid to camera placement, ensuring an optimal angle that encompasses all potential human movements, especially in a supine position. However, scenarios involving extreme rotational angles, such as when one sleeps with a hand covering the face, pose a limitation for detection, a constraint not easily overcome even by vision systems boasting a substantial camera array.The second challenge pertains to low-light environments, where adequate illumination becomes indispensable. In such instances, a system equipped with infera red becomes a requisite. Regrettably, the designed system does not incorporate ultraviolet lighting due to its impact on the sampling time, a compromise that was considered carefully during the system's development.

This points are added to discussion section:  In all image processing-based vision systems, two critical challenges commonly emerge. The first challenge revolves around concealed points, or "hidden points," denoting elements that elude the camera's line of sight due to their specific positioning angles. In this design, meticulous attention has been paid to camera placement, ensuring an optimal angle that encompasses all potential human movements, especially in a supine position. However, scenarios involving extreme rotational angles, such as when one sleeps with a hand covering the face, pose a limitation for detection, a constraint not easily overcome even by vision systems boasting a substantial camera array.The second challenge pertains to low-light environments, where adequate illumination becomes indispensable. In such instances, a system equipped with infera red becomes a requisite. Regrettably, the designed system does not incorporate ultraviolet lighting due to its impact on the sampling time, a compromise that was considered carefully during the system's development.

Q7_2: Thank you for your comment. As exemplified in Figure 1, the meticulously designed GUI is adorned with five discernible buttons, each endowed with specific functionalities. The "Browse" button, for instance, extends an invitation to users to traverse their computer's repository of recorded video clips in the ubiquitous mp4 format, facilitating the discernment and selection of target files or directories. Upon the invocation of the "Browse" button or option, a modal dialogue or folder dialogue elegantly materializes, thereby affording users the latitude to meticulously sift through and nominate the desired file or folder. Upon finalization of this selection process, the chosen pathway or file is ordinarily rendered manifest within a textual field or as an integral constituent of the broader user interface. Conversely, the "Start" button, when activated, triggers a well-defined course of action or process. In this context, the act of "starting" encompasses the initiation of a program and the instigation of a program workflow that serves the discerning purpose of delineating landmarks within the confines of the video clip. The "Pause" function, on the other hand, assumes the role of transiently suspending an ongoing process or task, ushering in a momentary hiatus to ensure precision and fine-tuning of proceedings. Moreover, the "Export Data" function assumes a pivotal function of both preserving and facilitating the seamless transference of data from the confines of the application to external files, seamlessly adopting the universally recognized CSV file format for maximal interoperability. Lastly, the "Quit" function, of monumental import, concludes the user's interaction with the application by orchestrating an elegant exit strategy. Prudent implementation of these multifaceted functions into the designed GUI inevitably culminates in a discernible augmentation of the application's usability and functional prowess. This augmentation fosters a heightened level of intuitiveness, thereby endearing the VSleep application to its user base and rendering it resoundingly user-friendly.

The following text was added to the main text after figure 1: As exemplified in Figure 1, the meticulously designed GUI is adorned with five discernible buttons, each endowed with specific functionalities. The "Browse" button, for instance, extends an invitation to users to traverse their computer's repository of recorded video clips in the ubiquitous mp4 format, facilitating the discernment and selection of target files or directories. Upon the invocation of the "Browse" button or option, a modal dialogue or folder dialogue elegantly materializes, thereby affording users the latitude to meticulously sift through and nominate the desired file or folder. Upon finalization of this selection process, the chosen pathway or file is ordinarily rendered manifest within a textual field or as an integral constituent of the broader user interface. Conversely, the "Start" button, when activated, triggers a well-defined course of action or process. In this context, the act of "starting" encompasses the initiation of a program and the instigation of a program workflow that serves the discerning purpose of delineating landmarks within the confines of the video clip. The "Pause" function, on the other hand, assumes the role of transiently suspending an ongoing process or task, ushering in a momentary hiatus to ensure precision and fine-tuning of proceedings. Moreover, the "Export Data" function assumes a pivotal function of both preserving and facilitating the seamless transference of data from the confines of the application to external files, seamlessly adopting the universally recognized CSV file format for maximal interoperability. Lastly, the "Quit" function, of monumental import, concludes the user's interaction with the application by orchestrating an elegant exit strategy. Prudent implementation of these multifaceted functions into the designed GUI inevitably culminates in a discernible augmentation of the application's usability and functional prowess. This augmentation fosters a heightened level of intuitiveness, thereby endearing the VSleep application to its user base and rendering it resoundingly user-friendly.

The authors would like to express their sincere appreciation to the reviewer for their thorough consideration of the manuscript and for providing valuable feedback, which undoubtedly enhanced its quality. Building upon these insightful comments, we have revised the article accordingly. We hope that the revised paper meets the expectations of the esteemed reviewer.

We extend our best wishes to you.

 Warm regards

Authors of  sensors-2619444
